# Characterization of a Clinically and Biologically Defined Subgroup of Patients with Autism Spectrum Disorder and Identification of a Tailored Combination Treatment

**DOI:** 10.3390/biomedicines12050991

**Published:** 2024-04-30

**Authors:** Laura Pérez-Cano, Luigi Boccuto, Francesco Sirci, Jose Manuel Hidalgo, Samuel Valentini, Mattia Bosio, Xavier Liogier D’Ardhuy, Cindy Skinner, Lauren Cascio, Sujata Srikanth, Kelly Jones, Caroline B. Buchanan, Steven A. Skinner, Baltazar Gomez-Mancilla, Jean-Marc Hyvelin, Emre Guney, Lynn Durham

**Affiliations:** 1Discovery and Data Science (DDS) Unit, STALICLA SL, Moll de Barcelona, s/n, Edif Este, 08039 Barcelona, Spain; francesco.sirci@stalicla.com (F.S.); jose.hidalgo@stalicla.com (J.M.H.); samuel.valentini@stalicla.com (S.V.); mattia.bosio85@gmail.com (M.B.); emre.guney@stalicla.com (E.G.); 2JC Self Research Institute, Greenwood Genetic Center, Greenwood, SC 29649, USA; lboccut@clemson.edu (L.B.); cskinner@ggc.org (C.S.); lncasci@clemson.edu (L.C.); sujatas@clemson.edu (S.S.); bkjones@clemson.edu (K.J.); cbuchanan@ggc.org (C.B.B.); sas@ggc.org (S.A.S.); 3Healthcare Genetics and Genomics, School of Nursing, Clemson University, Clemson, SC 29634, USA; 4Drug Development Unit (DDU), STALICLA SA, Avenue de Sécheron 15, 1202 Geneva, Switzerland; liogierx@gmail.com (X.L.D.); baltazar.gomez-mancilla@stalicla.com (B.G.-M.); jean-marc.hyvelin@stalicla.com (J.-M.H.); 5Research and Education in Disease Diagnosis and Interventions (REDDI) Lab, Center for Innovative Medical Devices and Sensors (CIMeDS), Clemson University, Clemson, SC 29634, USA; 6Department of Neurology and Neurosurgery, McGill University, Montreal, QC H3A 0G4, Canada

**Keywords:** precision medicine, ASD Phenotype 1, metabolic and transcriptomic alterations, NF-κB, NRF2, Warburg effect, cAMP, STP1 tailored treatment, DEPI

## Abstract

Autism spectrum disorder (ASD) is a heterogeneous group of neurodevelopmental disorders (NDDs) with a high unmet medical need. The diagnosis of ASD is currently based on behavior criteria, which overlooks the diversity of genetic, neurophysiological, and clinical manifestations. Failure to acknowledge such heterogeneity has hindered the development of efficient drug treatments for ASD and other NDDs. DEPI^®^ (Databased Endophenotyping Patient Identification) is a systems biology, multi-omics, and machine learning-driven platform enabling the identification of subgroups of patients with NDDs and the development of patient-tailored treatments. In this study, we provide evidence for the validation of a first clinically and biologically defined subgroup of patients with ASD identified by DEPI, ASD Phenotype 1 (ASD-Phen1). Among 313 screened patients with idiopathic ASD, the prevalence of ASD-Phen1 was observed to be ~24% in 84 patients who qualified to be enrolled in the study. Metabolic and transcriptomic alterations differentiating patients with ASD-Phen1 were consistent with an over-activation of NF-κB and NRF2 transcription factors, as predicted by DEPI. Finally, the suitability of STP1 combination treatment to revert such observed molecular alterations in patients with ASD-Phen1 was determined. Overall, our results support the development of precision medicine-based treatments for patients diagnosed with ASD.

## 1. Introduction

Autism spectrum disorder (ASD) is an etiologically and clinically heterogeneous group of neurodevelopmental disorders (NDDs). According to the Diagnostic and Statistical Manual of Mental Disorders, Fifth Edition (DSM-5) criteria [1], the diagnosis of ASD is based on core behavioral symptoms such as impaired social communication and repetitive and restrictive behaviors. Based on recent estimates in the United States and Europe, one out of 36 to 89 children aged around 8 years receives a diagnosis of ASD [2,3]. Patients diagnosed with ASD, like many other NDDs, show a variety of clinical manifestations and genomic alterations, hindering the development of treatments that are effective at the population level for individuals with ASD. Consequently, there is still a high unmet medical need for ASD.

Given the broad spectrum of molecular etiologies underlying patients diagnosed with ASD, it is not surprising that no clinically proven treatment addressing the core symptoms of ASD exists to date. The drug treatments currently approved by the FDA for use in ASD—risperidone and aripiprazole—address certain behavioral features like irritability rather than the core symptoms [4]. Existing clinical trials rely on patients diagnosed on the basis of behavioral assessments, often without qualitatively accounting for either the core symptoms or other non-behavioral comorbidities. Several compounds, such as memantine and sulforaphane have shown mixed results in some of these trials [5,6], although they failed to achieve a clear improvement across the whole population of patients. Accordingly, given the clinical, genetic, and molecular heterogeneity observed in ASD, several recent studies have aimed at characterizing disease subtypes in ASD, albeit relying mainly on behavioral data [7].

Earlier studies on prepubertal individuals with ASD investigated how morphometric features extracted from structural magnetic resonance imaging (MRI) [8] and distances between standardized facial landmarks describing facial morphology [9] vary across individuals with distinct cognitive and language skills. Along these lines, a study by Libero and colleagues suggested the existence of a subgroup of patients with ASD characterized by an enlarged head circumference in early childhood [10]. In another study, Henry et al. analyzed behavioral data from 47 preteen children diagnosed with ASD and 58 typically developing (TD) age-matched children to characterize potential cognitive subtypes within the ASD population [11]. The authors assessed the fulfillment of various tasks related to non-social information processing skills including spatial working memory, response inhibition, facial recognition and affect. Using the measures obtained from these tasks, they built a random forest model and hinted at the existence of subgroups of individuals among both ASD and TD groups with small but significant differences in the resting-state functional connectivity MRIs.

On the other hand, Doshi-Velez and colleagues [12] turned to electronic medical records of patients older than 15 years old to delineate potential systems-level clinical manifestations across patients with ASD beyond the neurobehavioral criteria from the DSM. They grouped individuals with ASD based on the co-occurrence of medical comorbidities using hierarchical clustering, identifying various patient subgroups that predominantly have seizures, gastrointestinal and auditory disorders, and psychiatric disorders such as episodic mood disorders, bipolar disorder, depression, anxiety, and conduct disorders. More recently, in addition to medical records, they have expanded their analysis using data from healthcare claims, familial whole-exome sequencing, and neurodevelopmental gene expression to characterize a mechanistically defined subgroup of patients with different clinical, genetic, and transcriptomic characteristics [13]. The results from these studies suggest the existence of ASD subgroups with distinct clinical and etiologic differences driven by different genetic and environmental contributions.

More recently, various bioinformatic and machine learning (ML)-based approaches have been explored to address the challenges posed by ASD clinical variability and genetic heterogeneity [14]. Among these, DEPI^®^ (Databased Endophenotyping Patient Identification) is a systems biology, multi-omics, and ML-driven platform designed for the identification of biologically enriched subgroups of patients with NDDs and the potential corresponding tailored treatments. DEPI integrates information on NDD-risk factors including genetic variants, differentially expressed genes in large-scale case-control studies, and comorbidities observed across patients with neurodevelopmental disorders and uses this information to identify pathway-level perturbations associated with clinical manifestations observed in patients with NDDs. In this study, we used the DEPI platform for the stratification of patients with ASD and identified a first clinically and biologically defined subgroup of patients with ASD—which we named ASD Phenotype 1 (ASD-Phen1). The ASD-Phen1 subgroup was predicted to be defined by a convergent molecular pathophysiology linked to an over-activation of NF-κB and NRF2 transcription factors and the presence of two related clinical signs and symptoms: an enlarged head circumference and aggravation of core ASD symptoms during episodes of infection and/or fever. By conducting an observational and bio-sampling study enrolling 84 patients with ASD at the Greenwood Genetic Center (GGC), we aimed to (i) clinically confirm the existence of this subgroup of patients; (ii) validate their convergence at the biological level based on NF-κB and NRF2 transcription factor dysregulations; and (iii) explore drug treatments including sulforaphane (to further validate the mechanistic hypothesis) and STP1 (STALICLA’s Therapeutic Package 1, a combination of a PDE4/3 inhibitor (ibudilast) and an NKCC1 antagonist (bumetanide)) as a corresponding tailored treatment identified by DEPI.

## 2. Materials and Methods

### 2.1. Study Participants and Patient Stratification

Clinical identification of ASD-Phen1 patients was achieved through a prospective clinical study conducted at the Greenwood Genetic Center (GGC, IRB approval N° Pro00079347). A screening of medical records was performed on 313 patients with idiopathic ASD in the GGC database to look for patients who qualified to be enrolled in the study. Inclusion criteria consisted of a diagnosis of ASD according to DSM-5, supported by either ADI-R or ADOS-2 and a well-documented head circumference value. Parents or caregivers provided written informed consent and participants provided written informed consent when possible. All the patients enrolled in the study were previously screened for ASD-associated genetic alterations. In particular, molecular tests excluded abnormalities in plasma amino acid levels, major chromosomal abnormalities, and pathogenic variants in genes associated with ASD: *ADNP*, *ALDH5A1*, *AMT*, *AP1S2*, *ARID1B*, *ARX*, *ATRX*, *BCKDK*, *BRAF*, *CACNA1C*, *CASK*, *CDKL5*, *CHD7*, *CHD8*, *CNTNAP2*, *CREBBP*, *CTNNB1*, *DHCR7*, *DYRK1A*, *EHMT1*, *FGD1*, *FMR1*, *FOLR1*, *FOXG1*, *FOXP1*, *FOXP2*, *GABRB3*, *SLC2A1*, *GRIN2B*, *HDAC8*, *HOXA1*, *HPRT1*, *KDM5C*, *KIRREL3*, *L1CAM*, *LAMC3*, *MBD5*, *MECP2*, *MED12*, *MEF2C*, *MID1*, *NHS*, *NIPBL*, *NLGN3*, *NLGN4X*, *NRXN1*, *NSD1*, *NTNG1*, *OPHN1*, *PAFAH1B1*, *PCDH19*, *PHF6*, *PNKP*, *PQBP1*, *PTCHD1*, *PTEN*, *PTPN11*, *RAB39B*, *RAD21*, *RAI1*, *RELN*, *SCN1A*, *SCN2A*, *SETBP1*, *SETD2*, *SHANK3*, *SLC9A6*, *SMC1A*, *SMC3*, *STXBP1*, *SYNE1*, *TBL1XR1*, *TBR1*, *TCF4*, *TMEM231*, *TMLHE*, *TSC1*, *TSC2*, *TUBA1A*, *UBE3A*, *UBE3C*, *VPS13B*, *ZEB2* [15]. ASD-Phen1 patients were then identified as per the two main criteria predicted by DEPI, namely a head circumference (HC) value above or equal to the 75th percentile within the first two years of life (HC ≥ 75), and a systematic aggravation of ASD behavioral symptoms, so-called flares, during episodes of immune challenges such as fever and infection events (e.g., acute inflammation). Blood samples from patients matching or not matching the criteria for ASD-Phen1—named ASD non-Phenotype 1 (ASD-non-Phen1)—were obtained for subsequent metabolic and transcriptomic profiling analyses (Appendix A). ASD-non-Phen1 patients were selected among patients with HC below the 75th percentile within the first two years of life in order to have participants with an even distribution of maximum head-circumference values, with ~1/3 having an HC below the 25th percentile, ~1/3 with HC between percentiles 25th and 50th, and ~1/3 with HC between percentiles 50th and 75th.

### 2.2. Participant-Derived Lymphoblastoid Cell-Line Generation

The generation of lymphoblastoid cell lines (LCLs) was attempted for all the ASD-Phen1 participants identified at the GGC and an equivalent number of ASD-non-Phen1 and typically developing (TD) individuals. Finally, LCLs were successfully generated from the following: ASD-Phen1 (N = 17), ASD-non-Phen1 (N = 19), and TD individuals (N = 20) (Appendix A). Peripheral blood samples were obtained by venipuncture and collected in tubes containing anticoagulant citrate dextrose (BD, Cat#364606 or 364816). The tubes were kept at room temperature and processed within 24 h. The blood was diluted with an equal volume of sterile Phosphate Buffer Solution (PBS, Corning, Corning, NY, USA), layered onto Ficoll (Ficoll-Paque™ Plus, Fisher, Hampton, NH, USA, Lot # 10255486) and centrifuged at 1000 rpm for 1 h. The mononuclear cells were collected, PBS was added, and the cell suspension was centrifuged at 1500 rpm for 5 min. The cells were re-suspended in 1 mL of Fetal Bovine Serum (Optima, Atlanta Biologicals, Flowery Branch, GA, USA, Lot # E17075) and 3 mL of RPMI medium (RPMI-1640, Corning). A 1 mL volume of titrated EBV was added, and cells were transferred to a culture dish containing phytohemagglutinin (Gibco-BRL, Fisher Scientific, Waltham, MA, USA). The culture medium composition was RPMI medium modified with 15% FBS, 1% L-glutamine (Corning) and 1% antibiotic/antimycotic (Corning), and it was incubated at 37 °C at 10% CO_2_. Cells were in culture for 5 to 7 days to ensure full transformation; after transformation was achieved, the cells were expanded. LCLs were harvested at −80 °C, and cell cultures in modified RMPI were performed from these stocks when required. Before the experiments, LCLs were harvested and centrifuged at 1000 rpm for 30 min, and pelleted cells were re-suspended in Biolog media.

### 2.3. Measurement of NADH Production in Participant-Derived LCLs

The Biolog Phenotype MicroArray Mammalian (PM-M) technology [16,17] was used to establish the energy metabolism profile of the different LCLs in the presence of diverse carbon energy sources (plates PM-M1 to PM-M4), as well as in the absence and in the presence of cyclic adenosine monophosphate (PM-M6, wells A1 to A12). LCLs (20 × 10^4^ cells per well) were incubated for 48 h at 37 °C in 5% CO_2_, using the modified Biolog IF-M1 medium (Biolog Inc., Hayward, CA, USA). Complete Biolog Media (Biolog IF-M1) was completed with 5.3 mL of FBS, 160 µL of L-glutamine and 1.1 mL of penicillin/streptomycin (Fisher Scientific), and was used for Biolog plates PM-M1 to PM-M4 (Biolog Inc.). In contrast, Biolog IF-M1 plus 5.5 mL of 100 mM glucose, 160 µL of L-glutamine, and 1.1 mL of penicillin/streptomycin was used for Biolog plate PM-M6. After the 48 h incubation period, Biolog Redox Dye Mix MB (Biolog Inc.) was added (10 μL/well) and the plates were incubated at 37 °C for an additional 24 h. As the cell metabolizes the carbon energy source, the tetrazolium dye in the media is reduced, producing a purple color according to the amount of NADH generated.

At the end of the 24 h incubation, the plates were analyzed utilizing a microplate reader (Tecan Spark, Männedorf, Switzerland) with readings at 590 and 750 nm. The first value (A_590_) indicated the highest absorbance peak of the redox dye, and the second value (A_750_) gave a measure of the background noise. The relative absorbance (A_590–750_) was calculated per well.

Similar experiments were repeated in the presence of sulforaphane 10 µM, a potent NRF2 activator, added to the culture medium prior to the addition of the Biolog Redox Dye Mix MB.

### 2.4. NADH Production Data Analysis

The NADH production of each group of LCLs (i.e., ASD-Phen1, ASD-non-Phen1, and TDs), following exposure to each carbon energy source in the control condition and in the presence of sulforaphane, from PM-M1 to PM-M4 plates, was analyzed against the negative control condition (i.e., blank plate without cells) using one-way analysis of variance (ANOVA) followed by Tukey’s post hoc test for multiple comparisons using GraphPad Prism8. Any compound with an absorbance value similar to levels found in the negative control conditions (i.e., background signal) was removed from further statistical analysis.

NADH production in the presence of various carbon sources was then compared between the 3 untreated-cell populations (ASD-Phen1, ASD-non-Phen1, and TDs LCLs) and between the 3 cell populations pre-treated with sulforaphane, by performing a principal component analysis with R software and the *prcomp* package.

The effect of db-cAMP on the NADH production in the presence of glucose was compared between the 3-cell populations through a One-way ANOVA followed by Tukey’s post hoc test for multiple comparisons.

### 2.5. RNA-Seq Profiling from Patient Blood Samples

RNA-seq analysis was conducted from blood samples obtained from 20 participants (randomly selected) from the GGC cohort: 10 ASD-Phen1 and 10 age-matched ASD-non-Phen1 participants (Appendix A). From each participant, two tubes each with 2.5 mL of whole blood in PAXGene^®^ RNA tubes (from PreAnalytiX, Hombrechtikon, Switzerland) were delivered and sequenced at Omega Bioservices facilities in Georgia (US). The RNA extraction was performed using the QIAGEN PAXgene Blood RNA Kit/Mag-Bind^®^ PX Blood RNA 96 Kit (QIAGEN, Hilden, Germany), and the ERCC Ex-fold RNA reagent (Thermo Fisher Scientific, Cat: 4456739, Waltham, MA, USA) was added to each sample. Depletion of rRNA and hgbRNA and the RNA-seq library preparation was performed using the Illumina TruSeq Stranded Total RNA with the Ribo-Zero Globin kit (Illumina, San Diego, CA, USA). The samples were then sequenced using the Illumina NovaSeq 6000 sequencer (Illumina) with a 2 × 150 bp configuration.

### 2.6. Differential Gene Expression Analysis

The two blood samples obtained from each patient were considered technical replicas. Quality control procedures were applied to search for the presence of strong batch effects among the samples by performing Principal Component Analyses (PCAs) using *prcomp* R function (*stats* package). Before collapsing replicates and running a differential expression analysis, a PCA plot was generated using expression values for all expressed genes within the cohort to assess the quality of the replicates and the existence of any prior clustering or batch effect. PCA in Appendix A shows that the samples from the same patient are grouped in pairs as technical replicates and that there is no evidence of strong clustering from subsets of samples, which would have indicated unannotated strong batch effects. A differential gene expression analysis between ASD-Phen1 and ASD-non-Phen1 was then conducted in order to characterize the ASD-Phen1-specific disease transcriptomic signature. The R DESeq2 package was used for the differential gene expression analysis, using the option *CollapseReplicates* and following the recommended setup for multiple sequencing runs from the same extracted RNA amount to increase statistical strength without introducing confounding batch effects. After the differential expression analysis, two additional PCA plots were produced to assess the power of differentially expressed genes to discriminate ASD-Phen1 vs. ASD-non-Phen1 individuals, first using differentially expressed genes with an adjusted *p*-value (according to the Benjamini and Hochberg correction method [18]) under 0.05, and, secondly, using the top 250 up-/down-regulated genes (sorted by Log2(FC)); this was carried out by considering only genes with median expression level across samples greater than 10 reads. The first two PCA coordinates were subsequentially used as features to train a logistic regression model to classify the two phenotypes using their transcriptomic profile. The models were validated using a leave-one-out strategy, where at each iteration a sample is classified using the others as a training set. The performances are computed as the number of correct classified samples in the whole dataset. To validate the procedure, we implemented a permutation test over 1000 iterations, where at each iteration phenotypes are randomly shuffled and the classification score is computed to obtain their null distribution.

Genes with an adjusted *p*-value smaller than 0.05 or ranked within the top 250 up-/down-regulated genes (sorted by Log2(FC)) were selected for downstream enrichment analyses. The threshold of the top 250 up-/down-regulated genes was selected based on the work of Iorio et al., 2010 [19].

### 2.7. RNA-Seq Profiling of Sulforaphane, STP1, and Vehicle-Treated Cell Lines

Sulforaphane, bumetanide, and ibudilast were purchased from MedChemExpress (Monmouth Junction, NJ, USA) as individual vials of 10 mM concentration in 1 mL DMSO and were of the highest purity available (99.75%, 99.95%, and 99.89%, respectively). STP1 was obtained by diluting and mixing bumetanide and ibudilast equally to obtain a final 1:1 ratio of the two individual compounds in the DMSO solution. Commercial A375 (ATCC CRL-1619), HT29 (ATCC HTB-38), LCL (ASTARTE BIOLOGICS 1038), NPC (ATCC ACS-5004), and MCF-7 (ATCC HTB-22) cell lines were purchased from the ATCC LGC Standards^®^ (Sesto San Giovanni, MI, Italy) and ASTARTE Biologics^®^ (Bothell, WA, USA) providers. The transcriptional signatures of sulforaphane and STP1 were obtained by treating commercial A375, HT29, LCL, NPC, and MCF-7 cell lines with a 5 μM final concentration of sulforaphane or control vehicle (DMSO) and MCF-7 and NPC with a 5 μM final concentration of STP1 or control vehicle (DMSO) for 6 and 24 h. The selection of these cell lines was based on the expression of target genes for sulforaphane and the STP1 individual compounds. In addition, LCLs derived from two ASD-Phen1 patients (STPT000247 and STPT000248) and a commercial (i.e., control) LCL cell line were randomly selected and treated with 5 μM of STP1 or DMSO for 48 h. Each cell line was cultured following the provider’s guidelines until a sufficient number of cells was reached to be seeded for treatment with sulforaphane, STP1, or DMSO. All the treatments were performed in triplicates, seeding 50,000 cells/well at the two time points. RNA was extracted from every single well, obtaining a total of 90 samples for library preparation and sequencing. Total RNA was quantified using the Qubit fluorometric assay (Thermo Fisher Scientific). Libraries were prepared from 125 ng of total RNA and sequenced on a NovaSeq 6000 sequencer in single-end mode with a read length of 75 bp and 4 M reads per sample. Bioinformatics analysis consisted of quality filtering, trimming, and alignment to the reference genome to generate raw data. The raw expression data were finally normalized and analyzed for each cell line. A final list of differentially expressed genes for each drug treatment vs. DSMO treatment was obtained across cell lines. In addition, a final list of genes differentially expressed in each DMSO-treated patient-derived LCL vs. DMSO-treated control LCL was also generated for each of the two ASD-Phen1 patients from which LCLs were derived. These two lists explored transcriptional alterations at the individual level in these two ASD-Phen1 patients. Similarly to Iorio et al. [19], a modified weighted-average merging method that considers the Spearman correlation across cell-type individual treatments was applied to generate drug-elicited transcriptomic responses for sulforaphane and STP1 by combining individual treatments across cell lines. Finally, a total of six different transcriptional signatures were obtained: (i) STP1 vs. DMSO treatment in MCF-7 and NPC; (ii) sulforaphane vs. DMSO treatment in A375, HT29, LCL, MCF-7, and NPC; (iii) STP1 vs. DMSO treatment in LCL derived from patient STPT000248; (iv) STP1 vs. DMSO treatment in LCL derived from patient STPT000247; (v) DMSO treatment in LCL derived from patient STPT000248 vs. DMSO treatment in control LCL; and (vi) DMSO treatment in LCL derived from patient STPT000247 vs. DMSO treatment in control LCL.

### 2.8. Gene Set Enrichment Analyses

EnrichR, a comprehensive gene set enrichment analysis web server [20,21,22] was used to explore the pathway enrichment of differentially expressed genes (adjusted *p*-value below 0.05) in ASD-Phen1 vs. ASD-non-Phen1. Gene sets and pathways in MsigDB Hallmark 2020, the KEGG 2021 Human, the Reactome 2022, and the WikiPathway 2021 Human databases were used in this analysis. Gene set enrichment analysis (GSEA) [23] was performed for the ASD-Phen1 vs. ASD-non-Phen1 differentially expressed genes with the aim of validating the consistency with NF-κB-related pathway enrichment among ASD-Phen1. The *fgsea* R package v1.10.1 was used to run the analysis [24]. A total of 15,497 gene ontologies (GOs) and canonical pathways (CPs) were considered in the analysis, as provided by the human MSigDB version 7.0 from the https://data.broadinstitute.org/gsea-msigdb/msigdb/release/7.0/ (accessed on 27 January 2021). A Fisher’s exact test was used to calculate the significance of an over-representation of NF-κB-related pathways (i.e., those implicating NF-κB) among those enriched (Benjamini–Hochberg adjusted *p*-value < 0.05), in the ASD-Phen1 vs. ASD-non-Phen1 transcriptomic signature.

The association between the ASD-Phen1 specific transcriptomic disease signature and an over-activation of the NF-κB and NRF2 transcription factors was then evaluated by assessing the enrichment (based on the two-sided Kolmogorov–Smirnov test) of NF-κB and NRF2 high-confidence targets [25] among the top 250 genes with the greatest up- or down-regulation (i.e., Log2(FC)) when comparing gene expression between ASD-Phen1 and ASD-non-Phen1 patients from the GGC cohort. The *ks.test* R function (*stats* package) was used to calculate the enrichment score (ES) and the associated significance (two-tailed test). Positive or negative ES values indicate that NRF2 gene targets tend to be up- or down-regulated in the transcriptomic signature.

The Kolmogorov–Smirnov test was then used to evaluate the effect of sulforaphane and STP1 on the activity of NF-κB and NRF2 transcription factors, measured by the expression of their targets in treated vs. untreated cell lines.

### 2.9. The Similarity between ASD-Phen1 and Relevant Drug Transcriptomic Signatures

The transcriptional similarity between the observed and reversed ASD-Phen1 disease transcriptomic signatures and 2100 drug candidates, including sulforaphane and STP1, was assessed using DEPI. A rank-based gene set ES [19,26] was used to identify drug candidates with the capacity to revert transcriptomic alterations defining complex disease signatures. The score also enables the identification of drugs that show high transcriptional similarity with the disease signature, and that might, therefore, trigger worsening effects in the corresponding patients. Genes in the observed ASD-Phen1 disease transcriptomic signature were ranked based on the observed Log2(FC), going from genes with the greatest up-regulation towards those with the greatest down-regulation, in comparison with ASD-non-Phen1. The reverse ASD-Phen1 disease signature was generated by inverting the gene ranks from the ASD-Phen1 disease signature.

## 3. Results

### 3.1. Clinical Validation and Prevalence of ASD-Phen1 Population

Screening of medical records was performed on 313 patients with idiopathic ASD, from the GGC database. The results revealed that 89 out of these 313 individuals (28.4%) had at least one well-documented measure of head circumference taken by a trained physician within the first two years of life. Among these 89 patients, 46 (51.7%) had at least one confirmed measure of head circumference above or equal to the 75th centile within the first two years of life (later referred to as HC ≥ 75).

The assessment of the second mandatory criterion (flares) was only possible in 41 out of 46 patients with HC > 75, with at least one reported well-characterized event of appreciable worsening of behavioral symptoms after episodes of immune challenges, including fever, vaccination, loss of a tooth, and other infection events (e.g., acute inflammation) in 20 of these patients. Thus, a total of 20 patients were identified as matching the inclusion criteria predicted by DEPI to define the subgroup called ASD-Phen1 among a total of 84 ASD patients for whom both criteria were possible to be assessed (i.e., excluding the five patients for whom the assessment of the second criterion was not possible). This corresponds to an overall observed prevalence of 23.8% for ASD-Phen1. The ASD-Phen1 subgroup demographic characteristics are summarized in Table 1 and described in more detail in Appendix A.

### 3.2. Metabolic Profile of LCLs from ASD-Phen1 versus ASD-Non-Phen1 and Control

The metabolic profile of LCLs successfully derived from ASD-Phen1 patients (N = 17) was characterized and compared to the metabolic profile of LCLs successfully derived from ASD-non-Phen1 (N = 19) and TD individuals (N = 20) (Appendix A). Using the PM-M1 assay of the Biolog PM-M platform, we assessed the ability of the LCLs to utilize the various carbon-based energy sources. The overall analysis of the metabolic profile was carried out through a principal component analysis, which revealed that the compounds accounting for the most variance observed in dimension 1 (Dim.1) were D-glucose-6-phosphate (DG6P), D-glucose-1-phosphate (DG1P), D-glucose, D-mannose, D-fructose-6-phosphate (D6FP), D-fructose, D-galactose, lactic acid, alpha keto glutaric acid, succinamic acid and, to a lower extent, mono-methyl succinate and L-glutamine (Figure 1A). The compounds accounting for the most variance observed in dimension 2 (Dim.2) were adenosine, inosine, and mono-methyl succinate (Figure 1A).

The distribution of TD and ASD-Phen1 LCLs partially overlapped along dimension 1 and were completely segregated along dimension 2 (Figure 1B), suggesting that the difference between the two phenotypes relates to their production of NADH following exposure to adenosine, inosine, or mono-methyl succinate. The distribution of ASD-non-Phen1 and ASD-Phen1 LCLs completely overlapped along dimension 1 and partially overlapped along dimension 2 (Figure 1B), suggesting that the difference between the two phenotypes also relates to their production of NADH following exposure to adenosine, inosine, or mono-methyl succinate.

To determine which metabolites caused a significant difference in the production of NADH from the different cell populations, we analyzed the absorbance results for each of these metabolites separately (Figure 2). LCLs from ASD-Phen1 patients produced significantly less NADH than LCLs from TD when exposed to DG6P, DG1P, D-glucose, D-mannose, DF6P, and D-galactose, and produced significantly more NADH than LCLs from TD when exposed to adenosine, inosine and mono-methyl succinate (Figure 2). LCLs from ASD-non-Phen1 patients produced significantly less NADH than LCLs from TD when exposed to DG1P, D-glucose, and DF6P (Figure 2).

### 3.3. Effect of Sulforaphane on the Metabolic Profile of LCLs from ASD-Phen1 versus ASD-Non-Phen1 and TD

Given that DEPI predicted an up-regulation of the NRF2 pathways in subjects with ASD-Phen1, we further investigated the potential dysregulation of NRF2 pathways and its impact on the bioenergetic profile of LCLs, by repeating a similar metabolic characterization of ASD-Phen1, ASD-non-Phen1, and TD LCLs pretreated with sulforaphane, a potent NRF2 activator.

Figure 3 shows the PCA analysis of LCLs pretreated with sulforaphane versus untreated LCLs. The distribution of sulforaphane-treated cells overlapped along dimension 1 and dimension 2, such that the clustering observed under control conditions (i.e., untreated LCLs) disappeared (Figure 3A), indicating that sulforaphane treatment abrogated metabolic differences between cell lines. Moreover, the distribution of ASD-Phen1 LCLs treated with sulforaphane overlapped with untreated ASD-Phen1 LCLs along dimension 2 and dimension 1. Interestingly, differences between ASD-Phen1- and ASD-non-Phen1-derived LCLs on NADH production in the presence of inosine also disappeared after treatment with sulforaphane (Figure 3B).

### 3.4. Over-Activation of NF-κB and NRF2 as the ASD-Phen1-Specific Disease Transcriptomic Signature

A differential gene expression analysis was conducted from blood samples from 10 ASD-Phen1 and 10 age-matched ASD-non-Phen1 patients. The twenty samples were selected with the purpose of having no significant differences in age between the two groups (*p* = 0.256; Appendix A). This analysis identified 10 genes with statistically significant differences in expression levels between ASD-Phen1 and ASD-non-Phen1 (adjusted *p*-value < 0.05; Table 2). The first two components of a PCA focused on this set of genes showed a clear separation between ASD-Phen1 and ASD-non-Phen1 patients (Figure 4A, highlighting a distinct gene expression signature in ASD-Phen1. When using the PCA coordinates to classify the sample’s phenotype, we recorded a perfect classification score for both when using the statistically significant genes and the most up/down-regulated genes. A permutation testing using a leave-one-out cross-validation confirmed the significance of the observed stratification as compared to the null distribution (*p* = 0.01).

To further characterize this signature, a gene set enrichment analysis was conducted using EnrichR (see Section 2), which mainly showed statistically significant enrichment (adjusted *p*-value < 0.05) of 6 out of these 10 genes on pathways related to the activation of NF-κB and downstream pro-inflammatory cascades, as well as on abnormal cell proliferation and de-differentiation processes (Appendix A). One of the six differentially expressed genes contributing to this enrichment is *PAX*5, whose haploinsufficiency has already been associated with risk of ASD and other neurodevelopmental disorders with high confidence [27,28,29]. In addition, *PNRC1* (also among these six genes) and two more genes also found to be differentially expressed in ASD-Phen1, namely *SWT1* and *TMA7*, have been previously suggested as candidate NDD-risk genes [30].

The separation between ASD-Phen1 and ASD-non-Phen1 patients is improved by expanding the PCA to the top 250 up- and down-regulated genes (Figure 4B), providing evidence of biologically sound signals beyond genes with statistically significant differences in their expression levels. Accordingly, a significant over-representation of NF-κB-related pathways was also observed among pathways significantly enriched (GSEA adjusted *p*-value < 0.05) for genes differentially expressed when considering the whole ASD-Phen1 vs. ASD-non-Phen1 transcriptomic signature (i.e., all genes; *p*-value = 7 × 10^−15^; 95% *conf. inter*. = 2.41–4.17; *odd ratio* = 3.19; Appendix A). We then further explored a putative over-activation of NF-κB and NRF2 transcription factors in ASD-Phen1, considering the distribution of the expression of NF-κB and NRF2 target genes in the whole ASD-Phen1 transcriptomic signature and in the LCL cell lines derived from two ASD-Phen1 patients by using a Kolmogorov–Smirnov test (see Section 2). Both NF-κB and NRF2 target genes were found to be significantly enriched (*p*-values < 1 × 10^−9^) towards up-regulated genes in the ASD-Phen1 group (Table 3). NF-κB targets were also significantly enriched towards up-regulated genes in the two individual ASD-Phen1 LCL-derived signatures (*p*-values = 0.01 and 1 × 10^−6^, respectively, Table 3). In addition, transcriptomic perturbations elicited by sulforaphane were found to be significantly similar to those observed in the ASD-Phen1 cohort (see Section 2) and in the two individual ASD-Phen1-derived LCL signatures. Sulforaphane was within the top drugs showing the highest transcriptional similarity for (i) the ASD-Phen1 cohort signature (Figure 5A; z-score = 2.257 and *p*-value = 0.02); (ii) the ASD-Phen1 patient STPT000248 signature (Figure 5B z-score = 3.09 and *p*-value = 0.002); and (iii) the ASD-Phen1 patient STPT000247 signature (Figure 5C; z-score = 2.46 and *p*-value = 0.01). Remarkably, as expected, the sulforaphane transcriptomic signature also shows significant enrichment (*p*-value = 5 × 10^−16^) towards up-regulation of NRF2 targets (Table 3). Still, it is significantly enriched (*p*-value = 2 × 10^−10^) for down-regulated NF-κB target genes (Table 3).

### 3.5. STP1 as a Suitable Drug Candidate to Revert NF-κB and NRF2 Over-Activation in ASD-Phen1

We used DEPI’s DGES module to identify drug candidates showing high transcriptional similarity with the reversed ASD-Phen1 gene expression signature (see Section 2), i.e., with the potential to revert the ASD-Phen1 transcriptomic alterations linked to increased NF-κB and NRF2 transcriptional activities. Among more than 2000 drug candidates explored, STP1, a combination of a PDE4/3 inhibitor and bumetanide, was found to be within the top 1% drug candidates with the reversed transcriptional perturbations (z-score = 2.878 and *p*-value = 0.004 for the reversed ASD-Phen1 cohort signature (Figure 5A); z-score = 3.090 and 2.512 and *p*-value = 0.002 and 0.01 for the reversed signatures from patients STPT000248 and STPT000247, respectively; (Figure 5B,C)). Importantly, the STP1-elicited gene expression profiles in both commercial and patient-derived cell lines show a significant enrichment towards down-regulation of both NRF2 and NF-κB targets, with a greater effect on NRF2 targets (*p*-values *=* 8 × 10^−4^, 2 × 10^−7^ and 4 × 10^−6^; Table 3).

### 3.6. Effect of Cyclic Adenosine Monophosphate on the Ability of ASD-Phen1 LCLs to Metabolize Glucose

One of the components of STP1—predicted by DEPI as a tailored-treatment candidate for the ASD-Phen1 subgroup—is an inhibitor of the cyclic nucleotide phosphodiesterase form 4 and form 3 (ibudilast), which leads to the accumulation of cAMP. In this context, we also assessed the effect of cAMP on the ability of the LCLs to metabolize glucose. The PM-M6 assay comprises, among others, six wells containing no compound and six wells with increasing concentrations of dibutyryl-cAMP (db-cAMP, a cell membrane-permeant analog of cAMP). Figure 6A shows the effect of increasing concentrations of db-cAMP on the NADH production by LCLs from ASD-Phen1, ASD-non-Phen1, and TD individuals. Whereas increased concentration of db-cAMP increases the production of NADH with a maximum reached for the two highest doses, increased concentrations of db-cAMP have no effect on LCL NADH production from ASD-non-Phen1 and TDs. Upon pre-treatment of LCLs from ASD-non-Phen1 and TDs with sulforaphane, increasing concentration of db-cAMP enhanced the NADH production, similarly to what is observed in untreated ASD-Phen1 LCLs. Interestingly, in LCLs from ASD-non-Phen1 and TDs pretreated with sulforaphane, increasing db-cAMP concentration induced a dose-related increase in the NADH production similar to what was observed in ASD-Phen1 LCLs (Figure 6B). These results indicate that upregulation of NRF2 in ASD-non-Phen1 and TDs mimics the specific cAMP sensitivity observed in LCLs from ASD-Phen1. Importantly, in LCLs from ASD-Phen1, pretreatment with ibudilast (the STP1’s PDE4/3 inhibitor component) showed a similar effect as compared to db-cAMP on the levels of produced NADH in the presence of glucose (Figure 6C).

## 4. Discussion

One of the main therapeutic challenges that remains in the field is to stratify patients with ASD to support the development of precision medicine-based therapies. While early ASD diagnostic markers have been gaining popularity in the past few years (e.g., eye tracking), they still disregard the heterogeneity across patients with ASD. In this study, metabolic and transcriptomic profiling of patients with ASD-Phen1 and ASD-non-Phen1, and of TD individuals revealed specific alterations in ASD-Phen1 patients as compared to ASD-non-Phen1 and/or TDs. More precisely, the energy production in ASD-Phen1, assessed by the production of NADH in patient-derived LCLs, was significantly increased in the presence of inosine, whereas it was decreased in the presence of DG6P, DG1P and D-glucose. Using an unsupervised PCA based on NADH production levels from 384 different energy sources, we observed a clear separation of ASD-Phen1 from ASD-non-Phen1 and TDs, especially along dimension 2. It is noteworthy that inosine is one of the carbon-based energy sources accounting for most of the variance observed along this dimension (Figure 1A). In anaerobic conditions, inosine is associated with increased energy production due to increased glycolytic activity [31]. Taken together, these metabolic differences suggest a metabolic reprogramming in ASD-Phen1 patients through a switch to glycolysis for ATP synthesis rather than the mitochondrial oxidative phosphorylation (regardless of the oxygen supply), coupled with an increase of the pentose phosphate pathway where the phosphorylated pentose moiety of nucleosides may be used as an energy source. This phenomenon has been previously described as the Warburg effect [32], and it is typically observed in cells with a high proliferating rate, such as cancer cells or activated lymphocytes. Following pre-treatment of LCLs with sulforaphane, the metabolic differences observed between ASD-Phen1, ASD-non-Phen1, and TDs disappeared and mimicked the untreated ASD-Phen1 LCLs profile, providing evidence for an over-activation of NRF2 transcription factor being involved in the metabolic reprogramming in ASD-Phen1. Similar metabolic reprogramming in tumoral cells has extensively been characterized by an up-regulation of NRF2 and NRF2-related pathways [33]. As previously mentioned, the ASD-Phen1 subgroup is characterized by an enlarged head circumference. Similarly to any other tissue, brain development is sustained by an intense cellular proliferation of stem cells. Fast-replicating cells acquire specific metabolic profiles to meet their high demands in energy but also their need for the synthesis of the various metabolites the cells need for their proliferation. The transition between proliferative neuronal progenitors and differentiated neurons will then require a metabolic reprogramming, from aerobic glycolysis to oxidative phosphorylation, that will allow the cells to switch from a fast production of small quantities of energy to a slower but more efficient production of ATP [34]. One of the master regulators of this metabolic shift is the mammalian target of Rapamycin (mTOR) [35], which together with PI3K and AKT pathways has been associated with enlarged head circumference in ASD [36] and is under the control of NRF2 [35,37].

Transcriptomic profiling of ASD-Phen1 and ASD-non-Phen1 patients provided additional evidence for an over-activation of NRF2 in ASD-Phen1, given the (i) significant enrichment of NRF2 targets on up-regulated genes in ASD-Phen1 patients (Table 3) and (ii) the similarity between differentially expressed genes in ASD-Phen1 patients and genes transcriptionally perturbed in cell lines pre-treated with sulforaphane (Figure 5). Interestingly, differential gene expression analysis also pointed towards an over-activation of NF-κB in ASD-Phen1, with significant enrichment of NF-κB target genes (Table 3) and genes involved in NF-κB activation pathways (Appendix A), which are found to be up-regulated in ASD-Phen1. A crosstalk between NF-κB and NRF2 has already been established [38]. Following inflammation, pro-inflammatory cytokines such as tumor necrosis factor (TNF)α, interleukin (IL)-1β, and bacterial lipopolysaccharide (LPS) activate NF-κB, which leads to the transcription of genes involved in inflammation development and progression; NF-κB can then up-regulate NRF2 [39,40], which can, in turn, inhibit NF-κB as a regulatory feedback loop [39] (Table 3). Imbalance between NRF2 and NF-κB pathways has been associated with several diseases, including autoimmune disorders and cancer [41]. Another possible factor affecting the metabolic and immune profile of these patients is their microbiota: there is vast evidence in the literature of abnormal frequencies of bacterial strains in children with ASD, as compared to controls, and microbiota transfer therapy seems to have lasting benefits for both gastrointestinal and behavioral symptoms [42,43]. However, abnormal representation of microbial strains may also result from a dysregulated immune activity, as the microbiota intertwines with other systems that are all influenced by the conjunction of the underlying genetic background/susceptibility and environmental factors. Nevertheless, microbiome analyses also open promising avenues for the future of biomarker discovery in NDDs [44].

Elevated serum levels of pro-inflammatory cytokines have been also previously reported in some patients diagnosed with ASD [45]. In ASD-Phen1, increased inflammatory responses could lead to constitutively active NF-κB and NRF2, which in turn could imply a metabolic reprogramming and the related increased cellular proliferation (Figure 7). Screening for drugs with the capacity to revert the transcriptional perturbations observed in ASD-Phen1 patients pointed to STP1, a combination of a PDE4/3 inhibitor (ibudilast) and an NKCC1 inhibitor (bumetanide), with demonstrated capacity to down-regulate both NRF2 and NF-κB in vitro (Table 3). Interestingly, increasing concentrations of cAMP lead not only to an increase in energy production in LCLs from ASD-Phen1 only, but also in LCLs from ASD-non-Phen1 and TDs pre-treated with sulforaphane, providing further evidence for the specificity of STP1 for ASD-Phen1 patients. Differences in cAMP cascade have been previously reported in LCLs from patients diagnosed with Fragile X syndrome (FXS), for which an imbalance in the NRF2 and NF-κB pathways triggered by an over-activation of RAC1 can also be hypothesized [39,46]. In patients with FXS, the defect in cAMP signaling [47,48] prompted clinical trials assessing the effect of a phosphodiesterase inhibitor. More precisely, preliminary clinical efficacy results using zatolmilast (BPN14770) were achieved in patients with FXS, showing an improvement in cognitive function, particularly language-related domains and daily functioning [49].

## 5. Conclusions

Altogether, our results provide evidence for the capacity of the DEPI platform to identify subgroups of biologically similar patients with ASD characterized by specific non-behavioral clinical signs and pathophysiological features. Moreover, they validate, for the first time, stratification approaches based on defined clinical comorbidities associated with ASD. With this observational and bio-sampling study we were able to (i) confirm the predicted existence of a subgroup of patients presenting the two clinical criteria at an observed prevalence ~24%; (ii) characterize metabolic and transcriptomic ASD-Phen1 specific signatures associated with over-activation of NRF2 and NF-κB transcription factors; and (iii) confirm in vitro the specificity of STP1 to revert such molecular alterations in patients matching the criteria for ASD-Phen1. Overall, our results support the use of integrative systems biology toward the characterization of mechanistically defined and clinically actionable subgroups in ASD and the corresponding tailored treatments. Lastly, this approach paves the way for precision medicine-based drug discovery in NDDs.

## 6. Patents

Some results described in this study were patented by L.D. and J.-H.H. (WO2020094748A1).

## Figures and Tables

**Figure 1 biomedicines-12-00991-f001:**
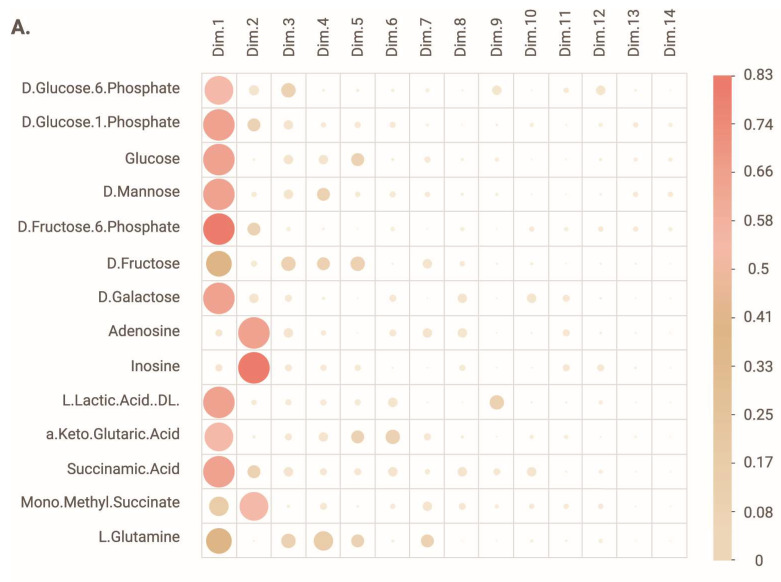
PCA analysis of absorbance measures at 24 h. (**A**) Correlation plot showing the contribution of each compound to the variance in dimension 1 and 2 (dim.1 and dim.2). Note that the bigger and darker the dot, the higher the contribution of the compound to the variance. (**B**) Biplot representation of the dim.1 and dim.2 with individual data points for each lymphoblastoid cell line (LCL), as well as the average ± s.d for each cohort. N = 20 LCLs from typically developing individuals (TD), N = 19 LCLs from ASD-non-Phen1 and N = 17 LCLs from ASD-Phen1.

**Figure 2 biomedicines-12-00991-f002:**
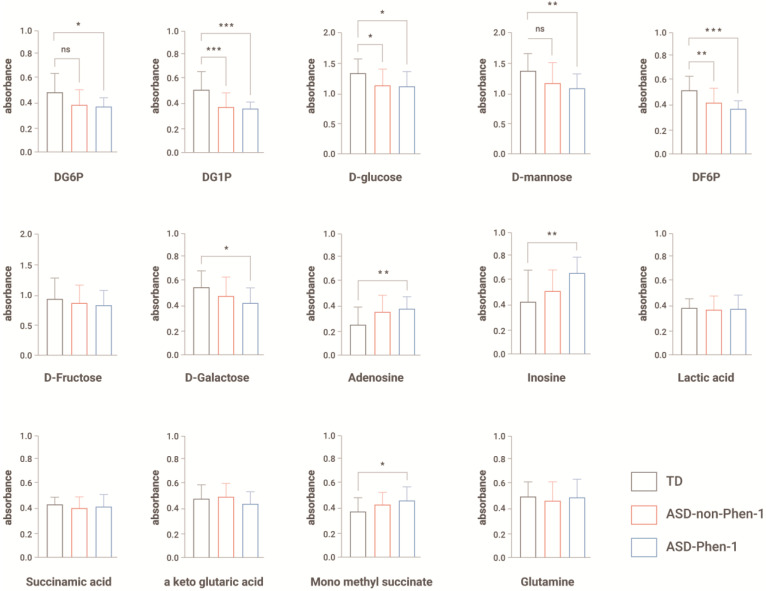
Absorbance measure after 24 h after exposure to each metabolite. Data are represented as average + s.d. One-way ANOVA followed by Tukey’s multiple comparisons. * *p* < 0.05, ** *p* < 0.01 and *** *p* < 0.001. N = 20 LCLs from typically developing individuals (TD), N = 19 LCLs from ASD-non-Phen1, and N = 17 LCLs from ASD-Phen1. ns: non-significant.

**Figure 3 biomedicines-12-00991-f003:**
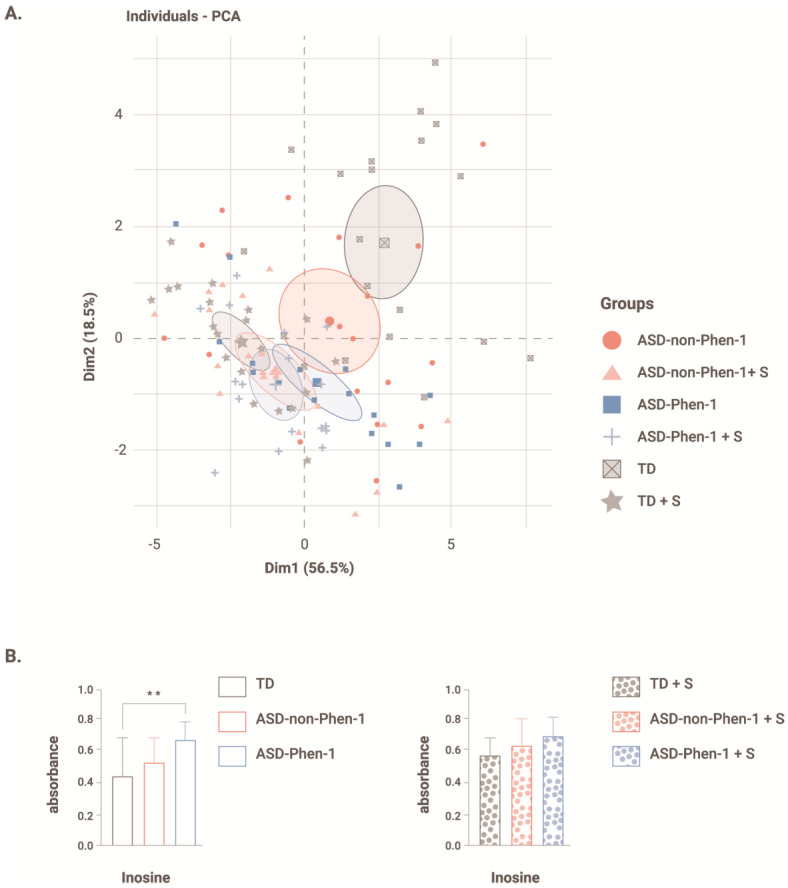
Effect of treatment with sulforaphane on PCA and absorbance after exposure to inosine. (**A**) Biplot representation of dimension 1 and dimension 2 with individual data points for each lymphoblastoid cell line (LCL) without treatment or treated with sulforaphane (+S), as well as the average ± s.d. for each cohort. N = 20 LCLs from typically developing (TD) individuals without and with pretreatment with sulforaphane (TD + S), N = 19 LCLs from ASD-non-Phen1 without and with pretreatment with sulforaphane (ASD-non-Phen1 + S), and N = 17 LCLs from ASD-Phen1 without and with pretreatment with sulforaphane (ASD-Phen1 + S). (**B**) Absorbance measured 24 h after exposure to inosine in LCLs from ASD-Phen1, ASD-non-Phen1 and TDs without treatment or treated with sulforaphane (+S). Data are represented as average + s.d. One-way ANOVA followed by Tukey’s multiple comparisons. ** *p* < 0.01. N = 20 LCLs from typically developing individuals (TD), N = 19 LCLs from ASD-non-Phen1 and N = 17 LCLs from ASD-Phen1.

**Figure 4 biomedicines-12-00991-f004:**
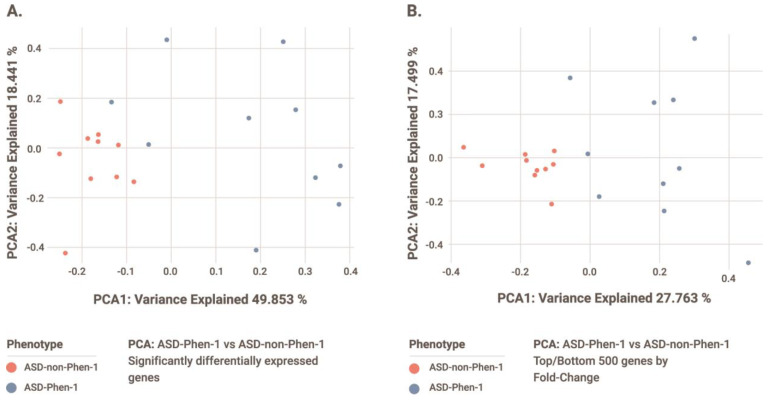
PCA of gene expression profiles from ASD-Phen-1 and ASD-non-Phen-1 patients using (**A**) statistically significantly differentially expressed genes and (**B**) top 250 up- and down-regulated genes (sorting by Log2(FC)) in ASD-Phen1 compared to ASD-non-Phen1.

**Figure 5 biomedicines-12-00991-f005:**
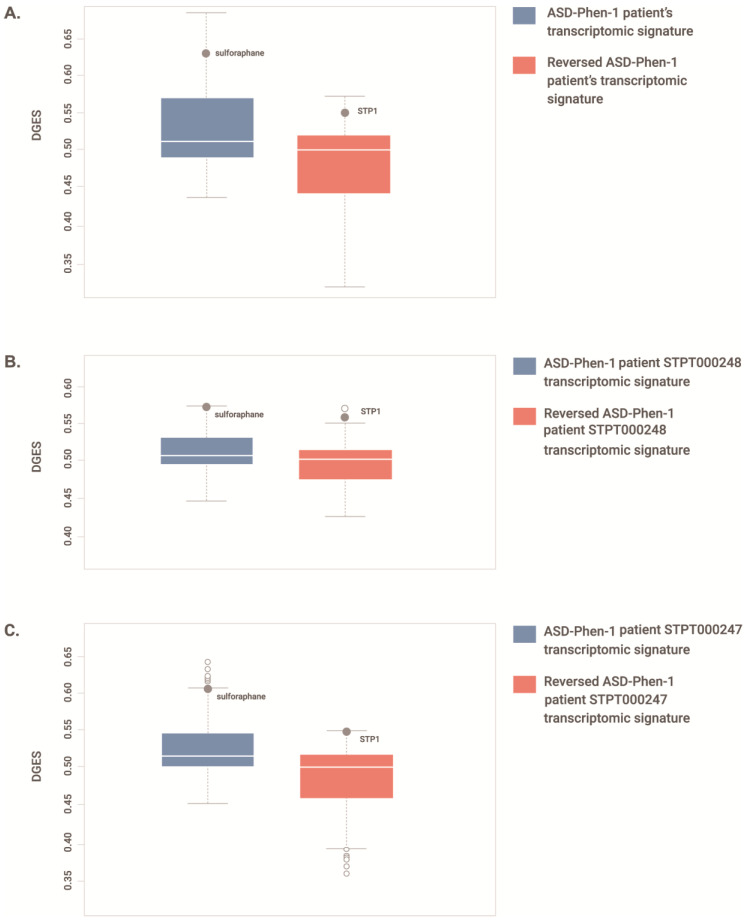
Distribution of transcriptional similarity (DGES) scores between drug and ASD-Phen1 gene expression profiles. DGES scores comparing a total of 2100 drug-elicited gene expression profiles. Boxplots show the first quartile (Q1/25th percentile): the middle number between the smallest number and the median of the dataset; the median (Q2/50th percentile): the middle value of the dataset; third quartile (Q3/75th percentile): the middle value between the median and the highest value of the dataset; interquartile range (IQR): 25th to the 75th percentile; maximum: Q3 + 1.5 × IQR; minimum: Q1 − 1.5 × IQR; outliers (open circles); position of the drug treatment (full circle). (**A**) the observed ASD-Phen1 vs. ASD-non-Phen1 disease transcriptomic signature (highlighting the position of sulforaphane) and the reversed ASD-Phen1 vs. ASD-non-Phen1 disease transcriptomic signature (highlighting the position of STP1); (**B**) the observed ASD-Phen1 STPT000248 vs. commercial LCL disease transcriptomic signature (highlighting the position of sulforaphane) and the reversed ASD-Phen1 patient STPT000248 vs. commercial LCL disease transcriptomic signature (highlighting the position of STP1); (**C**) the observed ASD-Phen1 patient STPT000247 vs. commercial LCL disease transcriptomic signature (highlighting the position of sulforaphane) and the reversed ASD-Phen1 patient STPT000247 vs. commercial LCL disease transcriptomic signature (highlighting the position of STP1).

**Figure 6 biomedicines-12-00991-f006:**
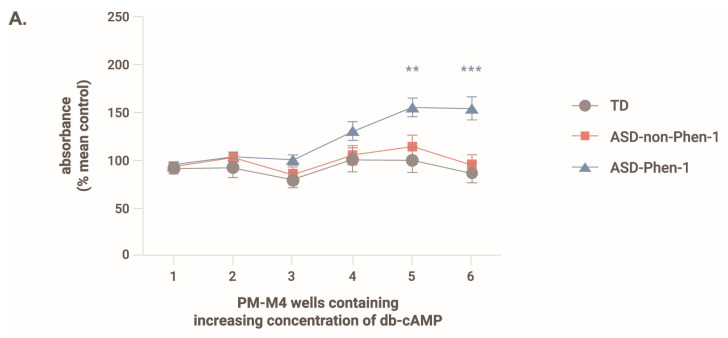
Absorbance measured 24 h after exposure to glucose and increasing concentrations of db-cAMP with/without sulforaphane. (**A**) absorbance in untreated LCLs, (**B**) absorbance in LCLs pretreated with sulforaphane. For each LCL, the absorbance value was normalized to that measured in the absence of db-cAMP. Data are presented as the average + s.d. One-way ANOVA followed by Tukey’s multiple comparisons. ** *p* < 0.01, *** *p* < 0.001. N = 20 LCLs from typically developing individuals without (TD) and with pretreatment with sulforaphane (TD + S), N = 19 LCLs from ASD-non-Phen1 without (ASD-non-Phen1) and with pretreatment with sulforaphane (ASD-non-Phen1 + S), and N = 17 LCLs from ASD-Phen1 without (ASD-Phen1) and with pretreatment with sulforaphane (ASD-Phen1 + S). (**C**) Absorbance value measured in LCL from ASD-Phen1 (N = 17) under baseline condition (i.e., without any metabolite effector) and in the presence of ibudilast (+ibudilast, 10 µM) or db-cAMP (well 6). **** *p* < 0.0001, ns: non-significant, one-way ANOVA.

**Figure 7 biomedicines-12-00991-f007:**
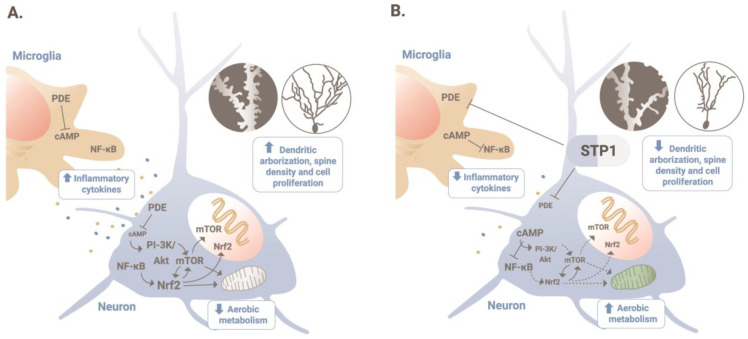
MoA of STP1 in ASD-Phen1. (**A**) Chronic increased inflammatory responses lead to a constitutively active NF-κB, and in turn, NRF2, in patients with ASD-Phen1; this results in an increased dendritic arborization, spine density, and cellular proliferation, as well as a decrease in the aerobic metabolism (also known as the Warburg effect). (**B**) By inhibiting PDE4/3 and, therefore, increasing cellular levels of cAMP, STP1 can inhibit NF-κB and decrease the activity of NRF2 to re-establish homeostatic cellular growth and metabolism.

**Table 1 biomedicines-12-00991-t001:** Summary description of demographic data of ASD-Phenotype1, ASD-non-Phenotype1 and typically developing study participants.

Group	Overall Gender Distribution (N Males/N Females)	Mean Age at Time of Blood Draw for LCLs * (y.) (s.d.)	Mean Age at Time of Blood Draw for RNA-Seq (y.) (s.d.)	Mean Head Circumference (HC) Percentile (s.d.)
ASD-Phen1	18/2	7 (5.99)	14 (7.53)	89 (6.99)
ASD-non-Phen1	18/1	6 (4.03)	9 (5.14)	32 (23.94)
TD	15/5	5 (1.57)	NA	NA

* This includes only the age of patients with LCLs used to generate Biolog data. NA: Not Applicable.

**Table 2 biomedicines-12-00991-t002:** Differentially expressed genes in ASD-Phen1 vs. ASD-non-Phen1 (adjusted *p*-value < 0.05). Ensembl, Entrez and HGNC gene IDs are provided. BaseMean is the mean value of expression (in absolute counts) for the gene in the population, and gene expression fold change is reported as the log2 of the expression fold change for the expression value for the ASD-Phen1 subpopulation when compared to ASD-nonPhen1. *p*-value and padj provide the statistical value and the multiple-test corrected statistical value of the differential expression.

Ensembl	Entrez	HGNC Symbol	baseMean	log2FC	*p*-Value	Padj
ENSG00000143546	6279	S100A8	2339	2.02	8.2 × 10^−8^	0.001
ENSG00000138738	11107	PRDM5	114	1.39	2.3 × 10^−7^	0.002
ENSG00000165480	221150	SKA3	151	1.64	3.3 × 10^−7^	0.002
ENSG00000109674	55247	NEIL3	423	1.54	4.5 × 10^−7^	0.002
ENSG00000116668	54823	SWT1	495	1.07	3.0 × 10^−6^	0.010
ENSG00000163221	6283	S100A12	587	1.67	4.1 × 10^−6^	0.012
ENSG00000196092	5079	PAX5	453	−1.02	1.3 × 10^−5^	0.030
ENSG00000232112	51372	TMA7	71	1.70	1.4 × 10^−5^	0.030
ENSG00000140379	597	BCL2A1	280	1.41	2.2 × 10^−5^	0.043
ENSG00000146278	10957	PNRC1	3133	0.44	2.5 × 10^−5^	0.044

**Table 3 biomedicines-12-00991-t003:** NRF2 and NF-κB gene-target enrichment analyses results for the different ASD-Phen1 transcriptomic signatures.

Transcriptomic Signatures	ESNRF2	*p*-Value	ESNF-κB	*p*-Value
ASD-Phen1 vs. nonPhen1	0.21	5.6 × 10^−9^	0.24	<1 × 10^−9^
patient STPT000248	−0.08	0.10	0.11	0.01
patient STPT000247	0.05	0.7	0.19	1 × 10^−6^
sulforaphane (5 µM)	0.30	5 × 10^−16^	−0.18	2 × 10^−10^
commercial LCL treated with STP1 (5 µM)	−0.13	8 × 10^−4^	−0.08	0.02
patient STPT000248 LCL treated with STP1 (5 µM)	−0.19	2 × 10^−7^	−0.15	7 × 10^−7^
patient STPT000247 LCL treated with STP1 (5 µM)	−0.17	4 × 10^−6^	−0.15	6 × 10^−7^

## Data Availability

The datasets used and/or analyzed during the current study are available from the corresponding author on reasonable request due to patient privacy constraints.

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
