# Peer review of "Characterization of a Clinically and Biologically Defined Subgroup of Patients with Autism Spectrum Disorder and Identification of a Tailored Combination Treatment"

_biomedicines, 2024, doi:10.3390/biomedicines12050991_

Round 1

Reviewer 1 Report

Comments and Suggestions for Authors

The authors aim to confirm the definition of a subgroup of ASD subjects obtained using (DEPI) which was based on a  Machine Learning analysis of clinical and biological characteristics.

Moreover, they tried to validate their convergence at the biological level based on NF-κB and 108 NRF2 transcription factor dysregulations; Finally, they evaluated specific drug treatments including sulforaphane, STP1, and an NKCC1 antagonist (bumetanide)) as tailored treatments.

Three groups of children have been taken into consideration:  ASD children characterised as phenotype 1 (identified by DEPI), ASD non phenotype 1, and children with typical development.

 The study is intriguing and the proposed goals are crucial for the study of ASD and for the management and targeted therapeutic approach.

However, some concerns emerge about the drafting and explanation of the results.

Major revision:

A supplementary table has been provided with punctual information of each case, however  a table with  demographic   description of (mean age, gender distribution), and clinical features (head circumferences , kind of acute   events .. ) should be added in the text to give a clear descrition  of subjects under study 

All the figures need to be improved in terms of clarity and explanation

An explanation of why gene expression has been evaluated only in 10 phen1 and 10 no phen1 ASD subjects, and no in  TD should be provided.

Minor revision

there is no clear definition of the start of the discussion section 

On the whole, there is no doubt that a significant study has been conducted, but the results and discussion need to be described more clearly.

Author Response

We thank the reviewer for the positive and constructive feedback. We hope the modifications in the revised version of the manuscript under the light of his/her comments now address the concerns below. Please see the attachment.

Reviewer 2 Report

Comments and Suggestions for Authors

To the AA

Idiopathic ASD affects a significant proportion of the pediatric population (1:89 to 1:36). Heterogenity is well-recognized in ASD. Moreover, the pharmacological treatments currently approved by the FDA in ASD – risperidone and aripiprazole – address behavioral features rather than the core symptoms. Hence, the development of precision medicine-based treatments in idiopathic ASD is an issue of key relevance. In the present study, by using a bioinformatic and ML-based approach, the AAs were able to identify a new subgroup of patients (named by the AAs as ASD-Phen1) in apx 1/4 (20/84=23.8%) of the whole examined idiopathic ASD population (n=313). Overactivation of NF-κB and NRF2 transcription factors related to two phenotypical signs (i.e., HC >75th pct and aggravation of core symptoms during episodes of infection - “flares”) were the distinctive features in this particular subgroup of patients.  A combination of the PDE4/3 inhibitor ibudilast and the NKCC1 inhibitor bumetanide (STP1) was predicted as a suitable drug candidate in reverting NF-κB and NRF2 over-activation. The overall data interpretation proposed by the AA is that a chronic inflammatory response in ASD-Phen1 patients leads to a constitutive activation of NF-κB and NRF2 in turn resulting in increased dendritic arborization -spine density -cellular proliferation, and reduced aerobic metabolism. Hence, inhibition of PDE4/3 - i.e. cAMP increasing cellular levels by STP1 is predicted to revert NF-κB- NRF2 activation and re-establish a physiologic homeostatic cellular growth and metabolism.

The Ms provides a fascinating insight, as well as a mechanistic hypothesis and prediction of candidate treatment for a subgroup of ASD patients representing about ¼ of the whole idiopathic ASD population. However, in my opinion, several issues should be addressed by the AA before considering the Ms suitable for publication.

Points

1.      Key clinical data should be provided in a table;

2.      A list of the genetic mutations explored and excluded in the examined idiopathic ASD patients should be added;

3.      Did the AA consider gender-related effects?

4.      Comments on potential effects of dysbiosis of the gut microbiota in ASD should be included in the discussion;

5.      Resolution quality of the figures needs significant improvement;

6.      Moderate language editing is needed.

Comments on the Quality of English Language

Moderate language editing is needed.

Author Response

We thank the reviewer for his/her constructive feedback and bringing up these pertinent points into our attention. We have addressed the reviewer’s concerns point by point in the attachment.

Round 2

Reviewer 1 Report

Comments and Suggestions for Authors

The manuscript has been adequately improved and the authors have sufficiently answered almost all my doubts

However the authors answered the specific request to add a summary description of demographic (mean age, gender distribution), and clinical features in the body of the text),  by adding details in the supplementary table.  Though it looks good, this action did not resolve my concern about a clear case study description overall.

Author Response

We thank the reviewer for the positive feedback and apologize for overseeing his/her concern in regard to the study group characteristics. We have now included a table in the main text with a summary description of demographic data, reflecting the overall case study description. We hope this modification does now address the concern.

Reviewer 2 Report

Comments and Suggestions for Authors

To the AA

I commend the AA for making major efforts in addressing the raised constructive criticism. The revised Ms is, in my personal opinion, significantly improved and potentially ready to be discussed in a larger scientific community readership.

Author Response

We thank the reviewer for appreciating the effort made towards addressing his/her raised concerns and for the positive feedback.